# Resistance to Plant Parasites in Tomato Is Induced by Soil Enrichment with Specific Bacterial and Fungal Rhizosphere Microbiome

**DOI:** 10.3390/ijms242015416

**Published:** 2023-10-21

**Authors:** Sergio Molinari, Paola Leonetti

**Affiliations:** Institute for Sustainable Plant Protection, IPSP-Bari Unit, Department of Biology, Agricultural and Food Sciences, DISBA, National Council of Research, CNR, 70126 Bari, Italy; paola.leonetti@ipsp.cnr.it

**Keywords:** PGPR, PGPM, plant immune system, priming, root microbiome, RKNs

## Abstract

Commercial formulations of beneficial microbes have been used to enrich the rhizosphere microbiome of tomato plants grown in pots located in a glasshouse. These plants have been subjected to attacks by soil-borne parasites, such as root-knot nematodes (RKNs), and herbivores, such as the miner insect *Tuta absoluta.* The development of both parasites and the symptoms of their parasitism were restricted in these plants with respect to plants left untreated. A mixture, named in the text as Myco, containing plant growth-promoting rhizobacteria (PGPR), opportunistic biocontrol fungi (BCF), and arbuscular mycorrhizal fungi (AMF) was more effective in limiting pest damage than a formulation containing the sole AMF (Ozor). Therefore, Myco-treated plants inoculated with RKNs were taken as a model for further studies. The PGPR contained in Myco were not able to reduce nematode infection; rather, they worsened symptoms in plants compared with those observed in untreated plants. Therefore, it was argued that both BCF and AMF were the microorganisms that colonized roots and stimulated the plant immune system against RKNs. Beneficial fungi colonized the roots by lowering the activities of the defense supporting enzymes endochitinases and β-1,3-glucanase. However, as early as three days after nematode inoculation, these enzyme activities and the expression of the encoding pathogenesis-related genes (*PR-2*, *PR-3*) were found to be enhanced in roots with respect to non-inoculated plants, thus indicating that plants had been primed against RKNs. The addition of paclobutrazol, which reduces salicylic acid (SA) levels in cells, and diphenyliodonium chloride, which inhibits superoxide generation, completely abolished the repressive effect of Myco on nematode infection. Inhibitors of copper enzymes and the alternative cyanide-resistant respiration did not significantly alter resistance induction by Myco. When Myco-treated plants were subjected to moderate water stress and inoculated with nematodes, they retained numbers of developed individuals in the roots similar to those present in regularly watered plants, in contrast to what occurred in roots of untreated stressed plants that hosted very few individuals because of poor nutrient availability.

## 1. Introduction

Chemical control of plant pests, particularly soil-borne parasites, has generally been recognized as harmful to human health and as contaminants to the environment, as well as ineffective in most cases; therefore, the use of biological tools to limit plant diseases and pests has been recommended [1]. Plant parasitic nematodes (PPNs) are the most diffused and damaging soil-borne pests, although nematode damage has largely been underestimated because the symptoms of infection are often non-specific and unrecognizable [2]. Among the many families of nematodes, the sole sedentary endoparasitic root-knot nematodes (RKNs), *Meloidogyne* spp., produce yield and quality losses that can be economically estimated in more than €80 billion/year in worldwide agriculture [3]. The invading motile worm-like juveniles (J2s) of RKNs penetrate the roots and move intercellularly through the cortex toward the central cylinder. They suck cell sap from a few cells of the central cylinder and inject, through a protrusible stylet of their mouth apparatus, several digestive compounds secreted by pharyngeal glands. J2s soon become sedentary and, through two molts to J3 and J4, develop into adult gravid females, which lay hundreds of eggs in gelatinous masses outside the roots. Nematode secretions alter gene regulation, cell metabolism, and structure as markedly as to transform the pierced cortical cells into giant nurse cells. Likewise, they produce hypertrophy and hyperplasia of the root tissue, resulting in the familiar visible galls or knots that are the only specific underground symptoms.

Biological control against RKNs implies the treatment of plants with specific rhizosphere microbes that can act as antagonists to nematodes in soil through antibiosis and competition for food and space and induce resistance by stimulating the plant immune system [4]. *Trichoderma* spp. and arbuscular mycorrhizal fungi (AMF) have extensively been reported to be effective resistance inducers against nematodes [5,6,7,8]. AMF contained in commercial formulations of biocontrol agents (BCAs) have been proven to colonize the roots of tomato plants and immunize plants against RKNs [9].

Some non-parasitic plant growth-promoting rhizobacteria (PGPR), such as *Pseudomonas fluorescens*, have been used as low-cost and environmentally safe BCAs and adopted in the management programs of soil-borne pathogens, including PPNs [10]. Competition for food and space, along with the production of metabolites effective in reducing egg hatching and juvenile vitality and induction of systemic resistance, are the main suppression mechanisms adopted by *P. fluorescens* to limit RKN infections [11,12]. Also, *Bacillus firmus*, another PGPR, has been used commercially as a bionematicide since the early 2000s, as it induces paralysis and mortality in individuals of several nematode families, RKNs included [13].

Such type of management of nematodes is in the frame of the recently proposed microbiome-mediated stress resistance in plants or, more generally, microbiome-assisted agriculture [14,15]. We have long been studying the potential of presently commercially available formulations containing a single or mixture of BCAs to control sedentary endoparasitic nematodes on vegetable crops in pots placed in a glasshouse [16]. The determination of the dosage expressed on a plant weight basis and also the period of time to allow endophytic fungi to colonize roots and immunize plants [9] has been found to be paramount in this management strategy.

In this study, we have enriched the soil of tomato-potted plants with the previously used mixture of bacterial and fungal rhizosphere microbes (named Myco in the text) or with an untested AMF commercial formulation (Ozor) and monitored the effects on plant growth, RKN infection, and on leaves attacked by the miner insect *Tuta absoluta*. As Myco proved to be more effective than Ozor in limiting pest parasitism, we focused on the mixture to find the microbial component(s) that specifically elicited plant resistance to RKNs. Moreover, we mixed several chemicals with Myco before providing microbial suspensions to plants in an attempt to characterize the biochemical pathway that endophytic fungi use to colonize roots and immunize plants. Immunization and immune reaction to RKNs were monitored by means of the expression of two pathogenesis-related genes (*PR* genes, such as *PR-2* and *PR-3*) and the activities of the encoded cell wall-degrading enzymes endochitinases and β-1,3-glucanase.

## 2. Results

### 2.1. Treatments with Rhizosphere Microorganisms to Test Nematode and Insect Infestation Levels in Tomato Plants

Tomato plants were soil-drenched with suspensions of commercial formulations, such as Myco, containing a mixture of rhizosphere microorganisms (AMF + BCF + PGPR), and Ozor, containing only AMF. Five days after treatments, the plants were inoculated with RKNs. Both treatments decreased nematode development in the roots with low plant fitness costs (Figure 1). Also, the amounts of egg masses produced per root system (EMs) were about 30% less in treated plants. However, females in treated plants had less competition for food, thus being able to lay more eggs per mass (female fecundity, FF, Appendix A). Therefore, reproduction potentials (RPs) of nematodes on treated plants were not significantly lower than those on untreated ones (Figure 1).

Myco treatment was also able to distinctly reduce infestation by the miner insect *Tuta absoluta* without affecting plant growth (Figure 2). Conversely, Ozor showed limited effectiveness.

When Myco and Ozor formulations were provided to plants in the absence of nematode inoculation, plant growth did not substantially differ with respect to untreated plants (Figure 3).

Since it appeared more efficient than Ozor in eliciting plant defense against nematodes and miner insects, Myco was chosen to investigate the family of rhizosphere microorganisms responsible for plant priming and immunity activation against the tested parasites. First, since Myco is a mixture of mainly AMF and PGPR, experiments were arranged to exclude one component from the mixture and test the sole AMF or PGPR as resistance effectors against RKNs. Amphotericin, a potent antifungal agent, only at the concentration of 1.7 mg g^−1^ Myco, was able to reverse the repressive effect of Myco on nematode infection, as this treatment significantly favored nematode development in the roots (Figure 4). It is likely that the PGPR present in Myco, when they are the only living microorganisms provided to plants, favor nematode infection. Conversely, when the sole AMF can act as resistance elicitors, as PGPR growth is repressed by antibiotics, induction of resistance is maintained.

As *Bacillus subtilis* was one of the PGPR contained in the Myco mixture that apparently favored RKN infection, suspensions of this rhizobacterium were tested for their effect on this pathogenesis [17]. Moreover, plants were soil-drenched with suspensions of *Pseudomonas fluorescens* ATCC 13525, which were generally reported to negatively affect RKN infections [18]. Data on plant growth and infection factors from treated and untreated plants are shown in Figure 5. *B. subtilis* was confirmed to worsen nematode infection factors; the higher the concentration of suspensions, the worse the infection (Figure 5B); roots of *B. subtilis*-treated plants showed a galling higher than untreated controls as to exceed their weight as a result of a more marked hypertrophy (Figure 5A).

Conversely, pre-treatments with *P. fluorescens* suspensions at the highest concentrations considerably repressed the development of nematodes in the roots (Figure 5E). However, considering the much lower number of individuals developing in the treated with respect to untreated roots and the consequent lower competition and higher availability of food, female fecundity was the highest in roots treated with 2.0 × 10^8^ CFU *Pf*/plant, thus increasing RP at values higher than those of controls (Figure 5F).

### 2.2. Molecular Mechanisms Associated with Myco-Induced Priming of Tomato Plants against RKNs

Once it was found out that the AMF/BCF contained in Myco were the most likely activators of plant priming against RKNs, some experiments were performed to investigate the molecular mechanisms supporting such priming. Therefore, the expressions of *PR-2–PR-3* genes, encoding β-1,3-glucanase and endochitinases, respectively, and their enzymatic activities were detected in roots 8 days after Myco treatments and compared with those from untreated roots. *PR-2* has been included in the SA-responsive *PR* genes associated with systemic acquired resistance (SAR), whilst *PR-3* over-expression was reported to be associated with resistance conferred by AMF to *M. incognita* in grapevine [19,20]. Expressions of *PR* genes did not significantly change at 8 dpt (days post-treatment) with Myco (Figure 6A); when the expression of the same genes was recorded at 3 dpi (days post-inoculation), a marked *PR-3* gene over-expression characterized roots of treated plants. Both endochitinases and β-1,3-glucanase activities were found to be inhibited in treated plants with respect to controls; however, if plants were inoculated with nematodes, such activities were significantly higher than those of control roots (Figure 6B,C).

### 2.3. Biochemical Effectors of Myco-Induced Priming of Tomato Plantsagainst RKNs

AMF contained in Myco have previously been reported to colonize tomato roots before exerting their protective effect against RKN infection [9]. Various chemicals were mixed into Myco suspensions before incubation and plant soil-drench to test if their presence would weaken or abolish the repressive effect of Myco, e.g., by compromising colonization of roots. The tested chemicals included SA, paclobutrazol (PCB), salicylhydroxamic acid (SHAM), diphenyliodonium chloride (DPI), and sodium diethyldithiocarbamate trihydrate (DIECA). SA is a SAR-eliciting phyto-hormone and induces Reactive Oxygen Species (ROS) accumulation in plants [21]. SHAM and DIECA are copper chelators, which are known inhibitors of the alternative cyanide-insensitive respiration of plants; SHAM is an H_2_O_2_ producer, and DIECA is a strong inhibitor of SOD and causes O_2_^●−^ accumulation [22]. PCB is an inhibitor of SA synthesis and is used to reduce SA levels [23], whilst DPI is a suicide inhibitor of NADPH oxidase and, consequently, O_2_^●−^ generation [24].

When Myco was incubated with PCB and the mixture provided to plants 5 days before RKN inoculation, the suppressive effect of Myco on nematode infection was completely abolished (Table 1), thus suggesting that SA is involved in AMF colonization of roots and/or in mycorrhizal-induced resistance (MIR) signaling. Accordingly, the mixture Myco + SA is able to further decrease infection symptoms despite the marked inhibition caused by the addition of the sole SA. Both DPI and SHAM inactivated the ability of Myco to act as a resistance elicitor, thus indicating that the Myco-repressive effect on RKN attack depends on O_2_^●−^ production and its dismutation into H_2_O_2_ catalyzed by SOD. When the sole DIECA was provided to tomato plants, nematode invasion of roots was largely repressed, probably because of the inhibition of cyanide-insensitive respiration, necessary for nematode metabolism, but also poor plant development due to DIECA toxicity. The association of DIECA with Myco probably reduced DIECA root uptake and relieved its deleterious effects on pests and plants.

Besides the results obtained on the combinations of Myco with the tested enzyme inhibitors, five different experiments shown in Table 1 confirmed the considerable inhibition of RKN infection consequent to Myco pretreatments; 77% SF and 59% EM reductions could be reached compared with non-pretreated plants. Again, such a reduction of the females developing in roots reduces competition for food and strikingly enhances female fecundity compared with females grown in untreated roots, sometimes doubling it, thus resulting in light or absent RP decrease.

### 2.4. Effect of Water Stress on Myco-Induced Priming against RKNs

The effect of Myco on nematode infection was tested in plants subjected to a period of moderate water stress. Before water stress induction, plants were treated with Myco and inoculated with RKNs after 5 days. Groups of untreated and treated plants were left unstressed as the control. At the end of such a period, stressed plants showed a loss of weight with respect to unstressed plants regardless of the fact that they had been treated before water stress induction with or without Myco (Figure 7A). The water stress condition negatively affected the development of sedentary forms in control roots, but it did not do so in Myco-treated roots (Figure 7B). On the contrary, reproduction rates did not change under water stress conditions in untreated roots, whilst it considerably increased in treated roots (Figure 7C).

## 3. Discussion

Enrichment for beneficial rhizosphere microbes can increase drought tolerance and stimulate the plant immune system, thereby promoting plant growth and crop yield [25,26]. However, beneficial rhizosphere microbes are naturally selected by plants or should be selected by farmers as an exogenously provided input to cropping systems to comply with the so-called microbiome-assisted agriculture [15]. In this study, we used a commercial mixture of beneficial rhizosphere microbes (named in the text as Myco) to test its effect on plant growth and protection against soil-borne parasites, such as RKNs, and foliar miner insects, such as *T. absoluta*. Treatments with Myco have actually already been reported to activate plant immune response and prime tomato plants against RKNs [6]. Moreover, it has been proven that, after Myco treatments on tomato plants, colonization of roots by AMF occurs, although colonization capacity, that is, the amount of root colonized in a certain time or the time taken to colonize roots, is strictly dependent on determined doses of inoculum [9]. Such doses were also important for the ability of treated roots to lessen RKN infection with respect to untreated roots. However, the role of each component of the mixture in the defense activation of tomato plants has not been fully cleared thus far.

Besides AMF as the main components, Myco contains antagonistic fungi, such as *Beauveria* spp. with entomopathogenic properties and *Trichoderma harzianum*. *Trichoderma* spp. cultivated in the laboratory or contained in commercial formulations were successfully used for biocontrol of RKNs [7,16]. Moreover, Myco contains PGPR, such as *Agrobacterium radiobacter*, *Bacillus subtilis*, and *Streptomyces* spp., which are known agents of biocontrol. First, we used another commercial formulation, named Ozor, which contains only AMF, to provide tomato plants as pre-treatments for immunization against RKNs. Ozor treatments were actually able to restrain nematode development in roots, compared with untreated roots, although not as much as Myco. The strong restriction of individuals developing in roots produced by these two commercially beneficial microbe formulations considerably lessens competition for food and enhances female fecundity, thereby allowing females to deposit a higher number of eggs in each mass than when under conditions of root high density. Therefore, reproduction rates sometimes are not significantly affected by beneficial microbe treatments. The higher attenuation of symptoms by Myco is also indicated by the healthier state of roots, compared with the highly infected control roots, indicated by lower weights. The difference between Myco and Ozor in the ability to control leaf mining by the miner insect *T. absoluta* was even larger. The mixture contained in Myco developed a root microbiome effective against aboveground insect pests, as already reported for other microbiomes against foliar pests and pathogens [27].

Although Myco was proven to be a good defense biostimulant, it did not act as a biofertilizer and growth promoter in unchallenged plants under the experimental conditions adopted in this study. In a previous study, plants were analyzed after much shorter periods of growth at which shoot weights were found to be enhanced in Myco-treated plants [9]. On the other hand, in plants infected by pests, the consistent relief of symptoms shown by Myco-treated attacked plants was not sufficient to make a difference in terms of plant weight growth, probably because of the fitness costs spent in immunity response that counterbalanced the growth-promoting effect exerted by Myco.

Once it was apparent that the mixture contained in Myco was more effective than Ozor, our task was to find which component of the mixture was specifically responsible for the effect. Amphotericin B, a potent antifungal agent, was added to Myco, and the mixture was incubated as usual before plant soil-drenching. When sufficient amounts of the agent were provided, Myco completely lost its property to lessen nematode infection; rather, its bacterial components increased root infection. On the contrary, antibiotics mixed with Myco did not alter its ability to restrain nematode development. Therefore, *B. subtilis*, a PGPR contained in Myco, was tested as a BCA against RKNs and compared with the PGPR *P. fluorescens*, which is known to control RKNs by the destruction of eggs and induction of plant defense mechanisms, leading to systemic resistance [28]. Treatments with *B. subtilis* increased infection indicators at any tested cell density; *P. fluorescens* was confirmed to effectively control nematode spread in the roots, at least at high suspension concentrations. Our findings seem to be in contrast to many studies in which *Bacillus* spp., due to their nematicidal properties, have generally been recognized as effective BCAs for RKNs [29]. However, such an inhibitory effect may depend on the species, and within *B. subtilis* sp., on different strains, and most probably on methods and concentrations of the application [30]. For comparison, we used the same low concentrations of *B. subtilis* suspensions per plant as those provided with Myco treatments, and they did not work. On the other hand, large amounts of studies have revealed that microbial-mediated modulation of host immune responses may facilitate nematode parasitism and that native microbiota naturally associated with nematodes, such as the one present in the fresh soil used in this study, may protect nematodes against microbial antagonists [31].

A series of experiments were undertaken to obtain insights into some possible molecular mechanisms underlying immune responses to nematodes induced by Myco treatments. In this study, we focused on the SA-responsive gene *PR-2*, which encodes for the enzyme β-1,3-glucanase, and on the JA-responsive gene *PR-3*, which encodes for endochitinases; both genes have recently been reported to trigger defense mechanisms against *Meloidogyne* spp. [32]. Both *PR-2* and *PR-3* were not responsive to Myco at least 8 days after treatments. However, the priming of plants was ascertained by the monitored over-expression of *PR-3* in plants treated with Myco 3 days after nematode inoculation. Moreover, in roots of inoculated Myco-treated plants, both chitinase and glucanase activities were higher than in untreated roots. Cell wall-degrading enzymes, such as chitinase and glucanase, are generally recognized to have an important role in the control of *Meloidogyne* spp. by opportunistic fungi, such as *Beauveria* and *Trichoderma* spp., mainly because of their capacity in inhibiting the hatching of eggs and inducing mortality of J2s [33]. Roots of resistant soybean cultivars showed enhancement of chitinase activity when attacked by *M. incognita* [34]. Augmented chitinase and glucanase activities observed in roots of Myco-treated plants 3 days after nematode inoculation can be considered as an early defense response induced by the interaction between roots and fungi present in the provided mixture. It was not a surprise to find that, if not inoculated by nematodes, Myco-treated roots conversely showed an inhibition of the tested cell wall-degrading enzyme activities with respect to control roots. Generally, non-pathogenic members of the root microbiome actively interfere with plant immune signaling to permit root colonization by secreting immune-suppressive effector molecules, like plant pathogens [35,36]. In particular, it was reported that SA-mediated defense was deactivated by *T. harzianum* endophytes up to 15 days after inoculation to favor their colonization of roots [7]. Secondary infections that may be carried out by soil-borne or above-ground plant parasites trigger immunity by overexpression of a series of genes involved in the defense response. Immune response activation is responsible for the low developmental rate of nematodes in immunized plants; however, such activation is preceded by root colonization of AMF and BCF present in the inoculum [6,9]. It is possible that AMF and BCF act together to contrast nematode attack, thus explaining the higher potential of Myco to limit the infection compared with Ozor.

To investigate which type of metabolic pathway Myco exerts its action, a series of metabolites and enzyme inhibitors were added to the treatments. Such compounds were mixed with Myco suspensions before the 3-day incubation; controls were arranged with suspensions of the sole compounds. Exogenously added SA has long since been proven to be an active inducer of resistance against RKNs [37]. The mixture of SA with Myco strengthened the repressive effect of SA on the development of SFs in roots, although it generally seems that mixing SA with Myco poses greater difficulties to the beneficials contained in Myco in their root colonization and consequent protective action. On the other hand, as biotrophs, the colonization of AMF is negatively impacted by SA, which is why they attempt to impair SA-mediated defense response [38]. However, when Myco was mixed with PCB, which reduced SA levels, it completely lost its defense-eliciting effect against RKNs, thus suggesting that, once roots have been colonized, AMF need suitable amounts of SA to trigger an immune reaction. It is also likely that MIR needs ROS generation since mixing Myco with DPI, which inhibits O_2_^●−^ generation, results in a loss of Myco protective effect against RKNs. On the contrary, DIECA is an O_2_^●−^ generator as it markedly inhibits SOD; DIECA limits plant growth and nematode development, almost impeding reproduction because it creates an oxidative environment in cells hostile to both host and pest. It is possible that Myco mixed with DIECA may lower DIECA absorption by roots, thus relieving its toxicity to plants and nematodes but weakening its own potential in colonizing and immunizing plants. SHAM reduces symptoms of infection, but in contrast to DIECA, it promotes plant growth and health since it inhibits the alternative cyanide-insensitive respiration that acts as a ROS scavenger [39], thus favoring the feeding and development of nematode juveniles. However, mixtures of Myco plus SHAM had effects on nematode infection similar to those obtained with the sole Myco, thus indicating that beneficials contained in Myco do not exploit plant cyanide-insensitive respiration and do not depend on copper enzyme activities to induce plant immune reaction. If it is MIR that Myco-treated plants use to contrast nematode attacks, it should mainly be mediated by SA signaling, according to our findings.

Among abiotic stresses, water stress is the most important growth-limiting factor, particularly in arid and semiarid regions. The role of beneficial PGPR and AMF in improving drought tolerance has recently been well described [40,41]. In our experimental conditions, though, Myco did not relieve the loss in plant weight caused by drought. Stressed roots hosted much fewer nematodes than unstressed roots because of food scarcity; this did not happen with Myco-treated roots, thus indicating that mycorrhizal roots save their capacity to feed a discrete number of parasites in spite of water stress. This finding indicates that mycorrhizal roots tolerate water stress and maintain their metabolic activities better than non-mycorrhizal roots, although this does not result in weight gain.

In nature, rhizospheres augment the number of microbes normally present in bulk soil and make a selection through specific root exudates for the most beneficial ones in terms of improved root architecture, nutrient uptake, abiotic stress tolerance, and faster and stronger immunity reaction [15]. Unfortunately, in most agricultural areas, the indigenous root microbiome has drastically been reduced by human agricultural practices or natural disturbances. Therefore, from the perspective of microbiome-assisted agriculture, large exogenous applications of beneficial microbes should be predicted. Data shown in this paper indicate some of the conditions needed to make sure these treatments may be effective and anticipate the problems that farmers will have to face when commercial formulations are going to be used. Experiments are being conducted to ascertain the durability of microbiome benefits in terms of protection against plant parasites when several crop seasons are considered. Finally, it is important to have information on the survival rates of the added microbe consortia in time and in competition with the indigenous ones present in the tested soils.

## 4. Materials and Methods

### 4.1. Treatment of Plants with a Mixture of Rhizosphere Microorganisms

Seeds of the tomato (*Solanum lycopersicum* L.) cultivars Roma VF, Regina, Fiaschetto, Principe Borghese, and Marmande, all susceptible to RKNs, were sown in special containers filled with sterilized topsoil at 23–25 °C in a glasshouse. Seedlings were transferred to 110 cm^3^ clay pots filled with freshly field-collected loamy soil and put in temperature-controlled benches (soil temperature 23–25°C). Seedlings were provided with a regular regime of 12 h light/day, fertilized weekly with Hoagland’s solution, and allowed to grow to a weight of 3–5 g. A mixture of rhizosphere microorganisms was provided to plants by soil-drenching in pots a commercial formulation (Micosat F^®^, named Myco in the text, C.C.S. Aosta, Italy) consisting of 25% arbuscular mycorrhizal forming fungi (AMF) provided as root fragments containing intra-radical spores/vesicles and hyphae of *Glomus* spp. (*Glomus* spp. GB 67, *G. mosseae* GP11, *G. viscosum* GC 41). Myco also contained PGPR, such as *Agrobacterium radiobacter* AR 39, *Bacillus subtilis* BA 41, and *Streptomyces* spp. (6.2%, 4.0 × 10^8^ C.F.U. g^−1^ Myco). Amounts of formulation were dissolved in sterile distilled water added with peptone –glucose (0.7 g L^−1^) and incubated in an orbital shaker at 25°C for 3 days in the dark. The amount in grams of Myco provided per plant was fixed according to the fresh weight of the plant at treatment (0.25 g Myco g^−1^ pfw) since this dose had previously been proven to prime tomato plants against nematodes with minimal fitness costs [6]. Since the formulation used contained both AMF and PGPR, some controls were made to investigate which components were responsible for plant priming. Therefore, in some experiments, Myco and control suspensions were added with 0.85 and 1.7 mg amphotericin B g^−1^ Myco, a potent antifungal compound, to exclude the fungal component from the soil-drenched formulation; conversely, minimal amounts (0.25 mg g^−1^ Myco) of antibiotics, such as ampicillin and streptomycin, were added to incubation media to exclude rhizobacteria.

To obtain information on the metabolic pathways through which Myco induces priming of tomato plants against RKNs, a series of chemicals was added to Myco and control suspensions before incubation. The chemicals are as follows:−salicylic acid (SA) as water solutions of potassium salicylate at 15–20 mg g^−1^ Myco−salicylhydroxamic acid (SHAM) as water solutions at 0.8 mg g^−1^ Myco−sodium diethyldithiocarbamate trihydrate (DIECA) as water solutions at 0.9 mg g^−1^ Myco−diphenyliodonium chloride (DPI) as water solutions at 0.3 mg g^−1^ Myco−paclobutrazol (PAC) as ethanol solutions at 3 mg g^−1^ Myco

In the cases of chemicals provided as ethanol solutions, the same amounts of ethanol were added to control suspensions.

The effect of moderate water stress on Myco-induced priming of plants was tested as well.

### 4.2. Treatment of Plants with AMF

Plants at the weight of 3–5 g were soil-drenched with the commercial formulation Ozor (Bioplanet, Italy), containing 500 propagules g^−1^ of *Glomus intraradices* CMCCROC7. Opportune amounts of the formulation were dissolved in sterile distilled water added with peptone–glucose (0.7 g L^−1^) and incubated in an orbital shaker at 25 °C for 3 days in the dark. The most effective dose was determined as 0.1 g Ozor g^−1^ pfw at treatment. Controls were provided with peptone–glucose suspensions incubated without Ozor.

### 4.3. Treatment of Plants with PGPR

*Pseudomonas fluorescens* ATCC 13525 and *Bacillus subtilis* ATCC 6633 were used as potential elicitors of resistance of tomato plants against RKNs. *P. fluorescens* is not contained in the mixture of BCAs provided as Myco, whilst *B. subtilis* is one of the components. *P. fluorescens* and *B. subtilis* were cultivated in shake cultures of King’s B medium and Agar Meat–Peptone, respectively, at 28 °C for 24 h in the dark. Bacterial suspension density was brought to an absorbance of 1.0, which is equivalent to a bacterial concentration of 1.2 × 10^9^ C.F.U. [42]. Aliquots of these mother suspensions were diluted with sterile water to soil-drench tomato plants with 0.5, 1.0, 2.0 × 10^8^ C.F.U./plant; control plants were provided with the same volumes of growing media.

### 4.4. Nematode Inoculation and Measurements of Infection Level

Field populations of the RKN *Meloidogyne incognita* (Kofi *et* White) Chitw. were reared on susceptible tomato plants in a glasshouse and used for inoculation of plants. Active second-stage juveniles (J2s) were obtained by incubation of egg masses in tap water at 25 °C in the dark. On the second day of incubation, J2s were collected and concentrated by filtering through 500 mesh sieves. J2s were counted by means of a dissecting microscope at 25× magnification. Aliquots of stirring suspensions containing approx. 400 J2s were poured into 2 holes made in the soil at the base of plants. Inoculations were made 5 days after Myco treatment. The described inoculations of susceptible tomato plants with field nematode populations gave severe standard infections.

Plants were grown under the conditions described above and harvested 2 months after inoculation. Under the adopted experimental conditions, most nematodes completed their life cycle with the deposition of eggs outside the roots in gelatinous egg masses (EMs) in about one month. Afterwards, J2s started to hatch in the potting soil, re-infesting roots as a second generation. The potential numbers of these second-generation J2s were much higher than those of the first artificial inoculation. However, in our experiments, second generation J2s entered the roots and developed into sedentary forms (SF: J3, J4, swollen females), but adult females did not have enough time to produce EMs for reproduction. Therefore, reproduction rates are those determined by the J2s of the artificial inoculation, whereas the total reported numbers of SFs developed in the roots are mostly indicative of the secondary infection. Reproduction potentials (RPs) indicate how many times the density of the initial population has multiplied after plant harvest, whereas SF numbers reflect the galling state of the roots; the higher the galling state, the higher the damage level of the plants.

At harvest, Myco-treated and control plants inoculated with or without nematodes, were weighed and their lengths measured. Then, roots, cut from shoots, were washed free of soil debris. Each sample for nematode life-stage extractions was recovered from 2 root systems that were chopped into fragments; samples were weighed before being used for extraction of SFs and eggs. Other weighed samples were used to determine the number of EMs. EMs were colored red by immersion of samples in 0.1 g L^−1^ Eosin Yellow and stored in a refrigerator for at least 1 h. Root fragments were scored for red-colored EMs under a stereoscope (6× magnification). Roots were softened by incubation in a mixture of the enzymes pectinase and cellulase at 37 °C in an orbital shaker before SF extraction to free sedentary nematodes from root tissue. Roots were then ground in a physiological solution, and SFs were collected on a 90 µm sieve. Aliquots (2 mL) of stirring nematode suspensions were pipetted in small Petri dishes, and the numbers of SFs were counted under a stereoscope (12× magnification). Eggs were extracted by stirring root samples in diluted bleach and counted under a stereoscope (25× magnification) [43].

If numbers of inoculated J2s are taken as the initial population in each pot (*P_i_*) and the number of total eggs per root system as the final population (*P_f,_* the number of J2s free in soil is negligible in the small pots used in the experiments),
RP is calculated as: RP = P_f_/P_i_
Female fecundity (FF) as: FF = eggs/EMs

### 4.5. Determination of Tuta Absoluta Infestation

Pots containing Myco-treated and untreated tomato plants were randomly located in a glasshouse naturally infested by *T. absoluta*. Twenty-one days later, the leaves of the plants were examined for the amounts of insect galleries. Infested leaves of each plant were collected, separated from healthy leaves, and counted. The number of insect galleries affecting each leaf was determinedas well.

### 4.6. Protein Extraction and Enzyme Activity Assays

Proteins were extracted from roots of plants harvested 8 days after Myco treatment and 3 days after nematode inoculation; as inoculation was carried out 5 days after Myco treatments, both plant groups, inoculated and not inoculated, had the same growth time. Root samples of all groups were collected, dried, weighed, and put on ice. Samples were ground in porcelain mortars by immersion in liquid nitrogen. Minced samples were further ground by a Polytron^®^ PT–10–35 (Kinematica GmbH, Malters, Switzerland) in 0.1 M KPi buffer (1:5 *w*/*v*, pH 6.0) containing polyvinylpyrrolidone (4%, PVP) and the protease inhibitor phenylmethanesulfonyl fluoride (PMSF, 1 mM). Coarse homogenates were filtered through four layers of gauze and centrifuged at 12,000× *g* for 15 min. In addition, 10mL syringes fixed with 0.45 µm nitrocellulose filters were used to further purify protein extracts. Proteins in the extracts were concentrated using ultra-filters (2mL Vivaspin micro-concentrators, 10,000 molecular weight cut off, Sartorius Stedim, Biotech GmbH, Rheinbrohl, Germany) by centrifugation (3000× *g*) at 4 °C. Retained protein suspensions were used for enzyme assays.

Chitinase activity (CHI) was detected as the amount of N-acetyl-D-glucosamine (NAG) produced from chitobiose by the β-glucuronidase introduced in the reaction mixture [44]. Root extracts (100 µL) were diluted in 150 µL of 0.05 M sodium acetate buffer (pH 5.2) containing 0.5 M NaCl and mixed with suspended chitin (250 µL, 10 mg/mL) from shrimp shells (Sigma-Aldrich, Milan, Italy). Reactions were started by incubating the mixtures in 1.5 mL Eppendorf tubes for 1 h at 37 °C in an orbital incubator and stopped by boiling at 100 °C for 5 min in a water bath. Then, reaction mixtures were centrifuged at 10,000× *g* for 5 min at room temperature, and 300 µL of supernatants were collected and added with 5 µL β-glucuronidase (Sigma, type HP-2S, 9.8 units/mL). After the start and end of reactions were performed as previously described, mixtures were cooled at room temperature, then again heated to 100 °C for 3 min after adding 60 µL of 0.8 M potassium tetraborate (pH 9.1) and cooled. Incubation at 37 °C for 20 min after the addition of 1% 4-dimethylamino-benzaldehyde (1.2 mL, DMAB, Sigma) was the next step. Absorbance was read at 585 nm (DU-70, Bechman), and the amount of NAG produced was determined by means of a standard curve obtained with known concentrations (4.5–90 nmoles) of commercial NAG (Sigma). Blanks (negative controls) were mixtures in which tissue extracts were not added; positive controls were arranged by adding 10 µL chitinase from *Streptomyces griseus* (Sigma, 200 units/g). Chitinase was expressed as nanokatal g^−1^ rfw, with 1 nkat defined as 1.0 nmol NAG produced per second at 37 °C.

β-1,3-Endoglucanase (glucanase, GLU) activity was measured by its capacity to release glucose from laminarin (Sigma, Italy). Reaction mixtures, incubated at 37 °C for 30 min, consisted of laminarin (0.4 mg) and 100 µL root extracts in 300 µL 0.1 M sodium acetate (pH 5.2). To test glucose production, Nelson alkaline copper reagent (300 µL) was added to the mixtures that were heated at 100 °C for 10 min and then cooled at room temperature. Nelson chromogenic reagent (100 µL) was added, and the absorbance of suspensions was read at 500 nm [45]. Negative and positive controls were arranged by substitution of root extracts with grinding buffer and the addition of laminarinase (2 U/mL), respectively. Enzyme activity was expressed as µmol glucose equivalents released per minute per g rfw according to a standard curve created with known amounts (10–200 µg mL^−1^) of commercial glucose (Sigma, Milan, Italy).

### 4.7. RNA Extraction, cDNA Synthesis, and Quantitative Real-Time Polymerase Chain Reaction

Roots from untreated and Myco-treated plants were collected and weighed at 8 dpt and 3 dpi by *M. incognita*. Root samples were ground to powder in a porcelain mortar in liquid nitrogen. Each total RNA extraction was carried out from 100 mg macerated tissue using an RNA-easy Plant Mini Kit (Qiagen, Germany) according to the instructions specified by the manufacturer. RNA quality was verified by electrophoresis runs on 1.0% agarose gel and quantified using a Nano-drop spectrophotometer. QuantiTect Reverse Transcription Kit (Qiagen, Germany) with random hexamers was used for cDNA synthesis from 1 μg of total RNA, according to the manufacturer’s instructions. PCR mixtures (20 μL final volume) contained RNAse-free water, 0.2 μM each of forward and reverse primers, 1.5μL cDNA template, and 10 μLSYBR^®^ Select Master Mix (Applied Biosystems, Italy). PCR cycling consisted of an initial denaturation step at 95 °C (10 min); 40 cycles at 95 °C (30 s), at 58 °C (30 s), and at 72 °C (30 s), with a final extension step at 60 °C (1 min). qRT-PCRs were performed in triplicate using an Applied Biosystems^®^ StepOne™ instrument. The following tomato genes were tested: *PR-2* (NM001247229, encoding β-1,3-glucanase) and *PR-3* (Z151140, encoding endochitinase). For each oligonucleotide set, a no-template water control was used. *Actin-7* (NM_001308447.1, *ACT-7*) was used as the reference gene for quantification, as its expression did not vary according to treatments. Sequences of primers for the genes are the following:

*PR-2* forward: AAGTATATAGCTGTTGGTAATGAA; reverse: ATTCTCATCAAACATGGCGAA

*PR-3* F: AACTATGGGCCATGTGGAAGA; R:GGCTTTGGGGATTGAGGAG

*Actin* F: GATACCTGCAGCTTCCATACC; R: GCTTTGCCGCATGCCATTCT

Gene transcript levels were expressed as 1/ΔC_t_, whereΔC_t_ is the difference between the cycle thresholds of fluorescence signal of the tested gene and the signal of the reference gene (*Actin 7*).

### 4.8. Experimental Design and Statistical Analysis

Data from plants treated with different rhizosphere microorganisms were confronted with those from untreated control plants. Data shown in each figure and related Appendix A are the means of values coming from 3 different bioassays; in each bioassay, the same treatment was applied to groups of 8 tomato plants. The means of plant growth factors were then calculated from 24 replicates. Conversely, one value of infection factors was obtained from 2 plants to obtain 4 replicates per experiment. The means of each factor are then calculated from 12 replicates. Conversely, in the experiments in which Myco is added with priming effectors, 2 bioassays were performed. The means of growth factors had 16, whilst those of infection factors had 8 replicates. Means ± standard deviations were separated by a paired *t*-test (Significance Levels: * *p* < 0.05 or ** *p* < 0.01) using MS Excel Software (version 2010).

For RNA extraction, plants coming from 2 independent bioassays were used; roots from 2 plants of the same group constituted one sample; RNA was extracted from 3 different samples of roots per treatment, harvested at 8 dpt and 3 dpi. qRT-PCR data are expressed as means (*n* = 6) ± standard deviations of 1/ΔC_t_ (Ct target gene–Ct actin). 1/ΔC_t_ means ± standard deviations were differentiated by a paired *t*-test (Significance Levels: * *p* < 0.05) using MS Excel Software.

## Figures and Tables

**Figure 1 ijms-24-15416-f001:**
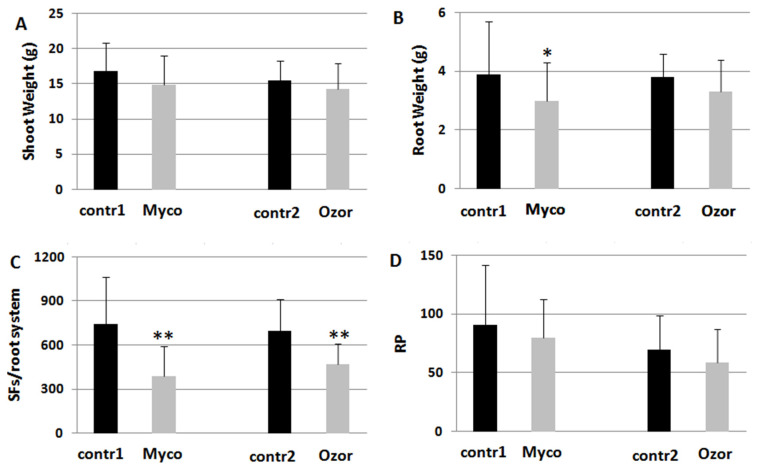
Plant growth (**A**,**B**) and infection indicators (**C**,**D**) of tomato plants harvested 2 months after inoculation with the root-knot nematode *M. incognita*. Treatments with 0.25 g Myco and 0.08 g Ozor g^−1^ plant fresh weight were carried out 5 days before nematode inoculation. Infection level was estimated by the number of sedentary forms (SFs) developed in each root system and by the reproduction potential (RP) of the nematode population. Means ± standard deviations for treated (Myco, Ozor) and untreated (contr1, contr2) plants were separated by a paired *t*-test (* *p* < 0.05; ** *p* < 0.01).

**Figure 2 ijms-24-15416-f002:**
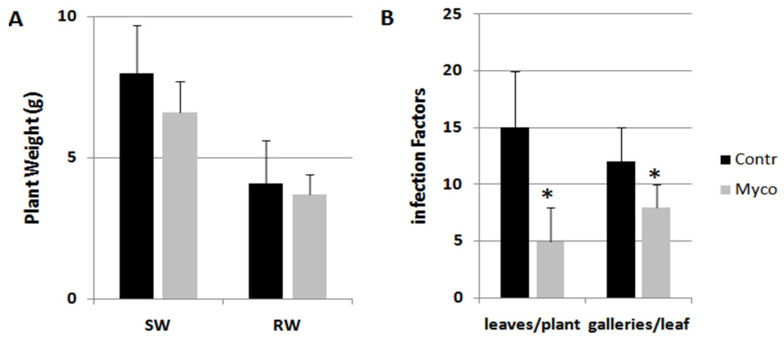
Tomato plants infested in the glasshouse by the miner insect *Tuta absoluta.* Treatments with 0.25 g Myco g^−1^ plant fresh weight were carried out 5 days before randomly locating plants in the infested glasshouse. Detected growth factors (**A**) were shoot (SW) and root weights (RW). Infection factors (**B**) were the number of leaves carrying at least one mine per plantand the number of galleries present on each infested leaf. Means ± standard deviations for treated (Myco) and untreated (Contr) plants were separated by a paired *t*-test (* *p* < 0.05).

**Figure 3 ijms-24-15416-f003:**
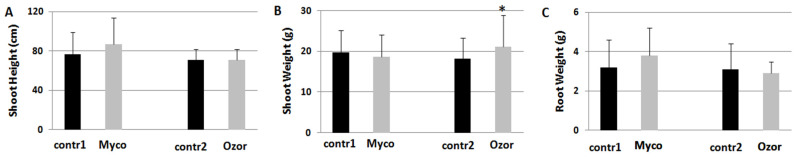
Plant growth expressed as shoot height (**A**) and shoot (**B**) and root (**C**) weights of tomato plants harvested 2 months after treatments with 0.25 g Myco and 0.08 g Ozor g^−1^ plant fresh weight. Means ± standard deviations for treated (Myco, Ozor) and untreated (contr1, contr2) plants were separated by a paired *t*-test (* *p* < 0.05).

**Figure 4 ijms-24-15416-f004:**
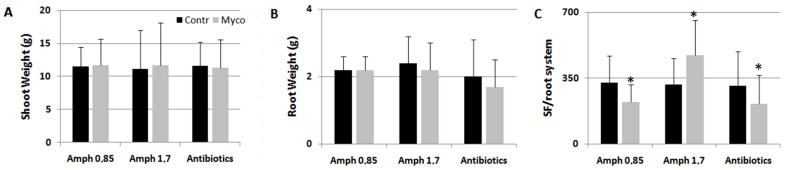
Tomato plants soil-drenched with Myco incubated with 0.85 and 1.7 mg amphotericin B g^−1^ Myco, a potent antifungal compound, and 0.25 mg g^−1^ Myco of antibiotics, such as ampicillin and streptomycin, added to exclude PGPR, were inoculated with RKNs. Shoot (**A**) and root (**B**) weights indicate plant growth rate; numbers of sedentary forms per root system (SF/root system, **C**) indicate the severity of infection. Means ± standard deviations for treated (Myco) and untreated (Contr) plants were separated by a paired *t*-test (* *p* < 0.05). Control plants were provided with suspensions containing amphotericin A and antibiotics without Myco.

**Figure 5 ijms-24-15416-f005:**
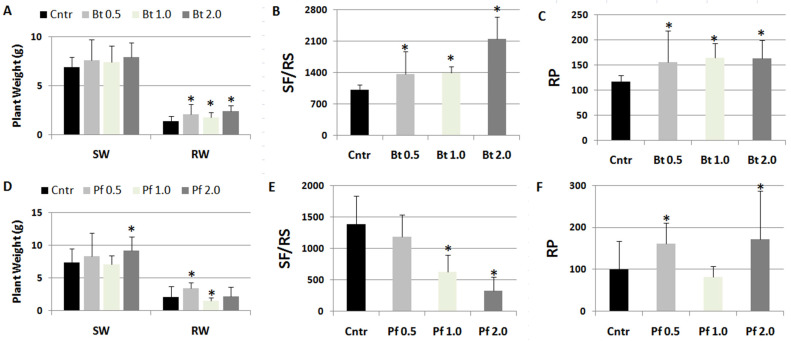
Tomato plants were soil-drenched with liquid suspensions (0.5, 1.0, 2.0 × 10^8^ CFU/plant) of *Bacillus subtilis* (Bt, **A**–**C**) and *Pseudomonas fluorescens* ATCC 13525 (Pf, **D**–**F**) and inoculated with *M. incognita* after 5 days. Two months after nematode inoculation, growth factors, such as shoot (SW) and root weights (RW), and infection factors, such as the number of sedentary forms per root system (SF/RS) and reproduction potential (RP), were detected. Means ± standard deviations for treated (Bt/Pf 0.5, 1.0, 2.0) and untreated (Cntr) plants were separated by a paired *t*-test (* *p* < 0.05).

**Figure 6 ijms-24-15416-f006:**
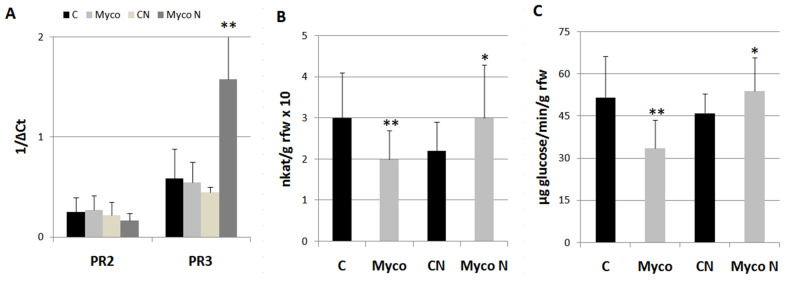
Expression of *PR-2* and *PR-3* genes in tomato roots (**A**), endochitinase activity (**B**), and β-1,3-glucanase activity (**C**) at 8 dpt (Myco) and a3 dpi with *M. incognita* (MycoN); untreated controls were inoculated (CN) or left uninoculated (**C**). Expression of genes was detected by quantitative real-time reversetranscription polymerase chain reaction (qRT-PCR). Gene transcript levels are expressed as 1/ΔC_t,_ where ΔC_t_ is the difference between the cycle thresholds of fluorescence signal of the tested gene and the signal of the reference gene (*Actin 7*). Endochitinase activity is expressed as nkat g^−1^ root fresh weight (rfw) with one nkat defined as 1.0 nmol NAG produced per second at 37 °C. β-1,3-glucanase activity is expressed as µmoles glucose min^−1^ g^−1^ rfw. Means ± standard deviations for treated (Myco) and untreated (**C**) plants were separated by a paired *t*-test (* *p* < 0.05; ** *p* < 0.01).

**Figure 7 ijms-24-15416-f007:**
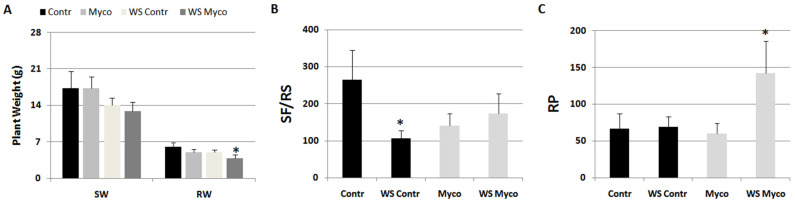
Plant growth (**A**), expressed as shoot (SW) and root (RW) weights, and infection indicators, expressed as sedentary forms per root system (**B**, SFs/RS) and reproduction potential (**C**, RP) of tomato plants harvested 2 months after inoculation with the root-knot nematode *M. incognita*. Groups of plants were subjected to moderate water stress (WS Contr); others were regularly watered (Contr). Before water stress induction, subgroups of both Contr and WS Contr were treated with Myco (Myco, WS Myco) and inoculated with nematodes after 5 days. Means ± standard deviations of each factor/indicator belonging to Contr/WS Contr and Myco/WS Myco subgroups were separated by a paired *t*-test (* *p* < 0.05).

**Table 1 ijms-24-15416-t001:** Growth factors, such as plant height (PH), shoot (SW) and root weights (RW), and infection indicators, such as sedentary forms per root system (SFs), reproduction potential (RP), egg masses per root system (EMs), and female fecundity (FF) detected in *M. incognita*-inoculated plants untreated (Cntr) and treated with Myco (+Myco). Control and Myco suspensions were also added before incubation with various biochemical effectors, such as PCB, SA, DPI, DIECA, and SHAM (3.0, 15–20, 0.3, 0.9, 0.8 mg g^−1^ Myco, respectively). Means ± standard deviations for treated (+Myco, Myco+Effector) and untreated (Cntr, Cntr + Effector) plants were separated by a paired *t*-test (* *p* < 0.05). The percentages of increase/decrease of significantly different factors/indicators are in parentheses.

	Cntr	+Myco	Cntr + PCB	Myco + PCB
PH	74 ± 22	82 ± 15	77 ± 25	83 ± 26
SW	18.0 ± 3.3	17.2 ± 2.5	19.1 ± 7.4	18.2 ± 3.5
RW	2.6 ± 0.8	2.8 ± 0.9	3.2 ± 1.4	3.3 ± 0.9
SFs	546 ± 286	333 ± 119 * (−39)	390 ± 256	482 ± 263 * (24)
RP	167 ± 50	116 ± 69 * (−31)	119 ± 40	101 ± 56
EMs	87 ± 13	52 ± 22 * (−40)	76 ± 26	79 ± 16
FF	543 ± 108	675 ± 231 * (24)	392 ± 213	424 ± 100
			**Cntr + SA**	**Myco + SA**
PH	81 ± 13	85 ± 10	81 ± 7	74 ± 15
SW	16.2 ± 2.7	16.6 ± 1.8	18.3 ± 3.1	16.2 ± 2.2
RW	4.3 ± 1.3	4.0 ± 1.1	4.0 ± 1.2	5.2 ± 0.8(30)
SFs	953 ± 286	222 ± 64 * (−77)	505 ± 152	372 ± 55 * (−26)
RP	60 ± 18	53 ± 15	38 ± 11	60 ± 9 * (58)
EMs	78 ± 23	32 ± 9 * (−59)	43 ± 13	47 ± 7
FF	222 ± 67	534 ± 154 * (140)	251 ± 75	415 ± 61 * (65)
			**Cntr + DPI**	**Myco + DPI**
PH	76 ± 25	73 ± 12	95 ± 17	71 ± 28 * (−25)
SW	17.0 ± 5.5	15.9 ± 3.2	23.4 ± 5.3	16.4 ± 8.2 * (−30)
RW	3.3 ± 0.8	2.1 ± 1.0 * (−36)	3.7 ± 1.3	2.6 ± 0.7 * (−30)
SFs	576 ± 154	211 ± 136 * (−63)	413 ± 181	501 ± 137 * (20)
RP	131 ± 39	109 ± 46	95 ± 52	81 ± 26
EMs	97 ± 35	48 ± 20 * (−51)	93 ± 27	86 ± 28
FF	535 ± 131	873 ± 503 * (63)	423 ± 154	387 ± 146
			**Cntr + DIECA**	**Myco + DIECA**
PH	110 ± 43	99 ± 20	81 ± 12	78 ± 13
SW	18.1 ± 4.9	18.3 ± 4.2	15.3 ± 6.8	14.8 ± 5.3
RW	2.5 ± 0.9	1.8 ± 0.6 * (−28)	1.8 ± 1.4	2.3 ± 1.9 * (28)
SFs	815 ± 284	290 ± 56 * (−64)	343 ± 258	588 ± 473 * (71)
RP	103 ± 36	69 ± 13 * (−33)	24 ± 18	69 ± 55 * (188)
EMs	138 ± 48	58 ± 11 * (−58)	37 ± 28	37 ± 30
FF	289 ± 101	455 ± 88 * (57)	250 ± 188	719 ± 578 * (188)
			**Cntr + SHAM**	**Myco + SHAM**
PH	83 ± 36	77 ± 41	92 ± 34	90 ± 36
SW	18.5 ± 8.0	19.4 ± 9.0	26.2 ± 8.3	28.1 ± 5.0
RW	3.6 ± 1.0	2.9 ± 1.9	4.4 ± 2.0	4.0 ± 1.7
SFs	597 ± 182	339 ± 280 * (−43)	434 ± 158	376 ± 239
RP	152 ± 55	149 ± 97	119 ± 59	128 ± 72
EMs	63 ± 17	44 ± 26 * (−30)	46 ± 29	55 ± 18
FF	539 ± 209	707 ± 491 * (31)	480 ± 206	609 ± 378 * (27)

## Data Availability

Not applicable.

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
