# Peer review of "Resistance to Plant Parasites in Tomato Is Induced by Soil Enrichment with Specific Bacterial and Fungal Rhizosphere Microbiome"

_ijms, 2023, doi:10.3390/ijms242015416_

Round 1

Reviewer 1 Report

The study titled " Resistance to plant parasites in tomato is induced by soil enrichment with specific bacterial and fungal rhizosphere microbiome" was conducted with a well-established concept of microbial consortium research and provides an overview of the beneficial microbes used to enhance the rhizosphere microbiome of tomato plants grown in pots in a glasshouse. Although the work is quite general, researchers have reported extensive information on the performance of inocula in crops. However, the manuscript has some merits, but it requires clarification on several critical points before it can be resubmitted for review and potential revision. The use of specific terms in the manuscript to highlight its idea is insufficient and more information is needed to fully understand the study's significance.

Minor comments:

1. I strongly suggest you that spell out all abbreviations in the text the first time mentioned in the text.

2. Cross-reference all of the citations in the text with the references in the reference section and make sure that all references have a corresponding citation within the text and vice versa.

3. Rectify all grammatical errors and line no missing in the manuscript, must add line number for specific comments in the next revision.

Major comments:

1. Need proper justification for the term “microbiome” “Myco” “Ozor”….

2. “Myco” – “consortium” and you can use “rhizosphere microorganisms”

Abstract:

3. It is advised that you rephrase the abstract, as it is quite lengthy and mainly focuses on introducing the idea, rather than highlighting the findings of the study.

4. Provide the detailed information of Bacillus subtilis; Bacillus subtilis (NCBI Accession no)

Keywords:

5. PGPR – PGPM; delete - AMF; BCF

Introduction:

6. There is a lot of information available on microbial consortia, which consist of bacteria, fungi, and other microorganisms such as arbuscular mycorrhizal fungi (AMF), and it is recommended to draw from this body of research when relevant. It is not necessary to cite or repeat previously published information.

Materials and Methods:

7. “Solanum lycopersicum L.” “Glomus intraradices” “Pseudomonas fluorescens”” Glomus spp.”…. – Italic…check full manuscript.

8.  Study area, collection of samples: add geographic coordinate.

9. Could you please provide a photograph of the experimental area, including some of the best-treated plants?

10. Please check: “SYBR® Select Master Mix” manufactured by Applied Biosystems, Italy, or  company?.

11. Provide the primer sequence details.

Results

12. Provide details company of commercial formulation of Myco.

13. Figure 1. “contr” - ?. write full or untreated

14.  Where Figure 2.?

15. What is the reason for growth differences in the treatment and control?

16. I recommend making a combined analysis and representation to better understand the findings.

17. The use of numerous abbreviations in the manuscript hindered readability and disrupted the flow of the text, which was a major issue.

Discussion

18. Scientific names in italics.

19. I recommend that you incorporate more detailed papers into your critical discussion, as they will help to better understand the scope of your idea.

Rectify all grammatical errors. 

Author Response

see the uploaded file

Reviewer 2 Report

In this study, the authors focused on Myco, a mixture of bacterial and fungal rhizosphere microbes, and monitored its effects on plant growth, RKN infection, and leaves attacked by the miner insect Tuta absoluta on tomato, as well as several chemicals that affect the efficiency of Myco. My comments/suggestions are as follows:

1. The abstract is too long and some information should move to the introduction part.

2. It would be better to show the infected symptoms in Figure 1.

In Figure 1, all the figures have two controls (for Myco and Ozor), they were both labeled as “contr” while they are different, they should be differentially marked, e.g.”contr 1” and “contr 2”.

What is the population of females in Figure 1?

3. It seems that Figure 2 is missing in the manuscript.

4. Fig. 3 shows the effects of Myco and Ozoron plant growth, which I think is most important and should be the first to test before all other experiments. As if it is not safe for the plant, the efficiency is meaningless.

5 It seems paradoxical that in Fig. E and F, under the treatment of Pf 2.0, the SF/RS significantly decreased but RP increased.

6. “when the expression of the same genes was recorded at 3 dpi (days post inoculation), a marked PR-3 gene over-expression characterized roots of treated plants.” this description is not consistent with Figure 6A.

Why did not the authors use the 2−∆∆CT method to calculate the relative expression?

7. Almost all the bacteria, fungi, and plant names that should be in italic but not.

“For comparison, we used the same low concentrations of B. subtilis suspensions per plant as those provided with Myco treatments and they did nor work.” I think the author want to say “did not work”

Why are the decimal points in all the Figures shown as comma?

King’s B medium and Agar Meat-Peptone, “1/ΔCt” should have proper quotations

Author Response

see the uploaded file

Round 2

Reviewer 1 Report

The manuscript has been significantly improved. I would strongly recommend the author read your paper one more time critically and amend the reviewer's comments in the revised manuscript.  Pg. 9.......: Scientific names - italicize (full manuscript)

NA

Reviewer 2 Report

Thanks for answering my questions. In Q2, I meant it would be better to show the pictures of plant conditions together with the graphs in Figure 1, but it is okay if the authors did not take any.

I have no more questions.